# Invariant and Transportable Representations for Anti-Causal Domain Shifts

Yibo Jiang[1] and Victor Veitch[2,3]

[1]*Department of Computer Science, University of Chicago*
[2]*Department of Statistics, University of Chicago*
[3]*Google Research*

## Abstract

Real-world classification problems must contend with domain shift, the (potential) mismatch between the domain where a model is deployed and the domain(s) where the training data was gathered. Methods to handle such problems must specify what structure is common between the domains and what varies. A natural assumption is that causal (structural) relationships are invariant in all domains. Then, it is tempting to learn a predictor for label $Y$ that depends only on its causal parents. However, many real-world problems are "anti-causal" in the sense that $Y$ is a cause of the covariates $X$—in this case, $Y$ has no causal parents and the naive causal invariance is useless. In this paper, we study representation learning under a particular notion of domain shift that both respects causal invariance and that naturally handles the "anti-causal" structure. We show how to leverage the shared causal structure of the domains to learn a representation that both admits an invariant predictor and that also allows fast adaptation in new domains. The key is to translate causal assumptions into learning principles that disentangle "invariant" and "non-stable" features. Experiments on both synthetic and real-world data demonstrate the effectiveness of the proposed learning algorithm. Code is available at https://github.com/ybjiaang/ACTIR.

## 1 Introduction

This paper concerns the problem of domain shift in supervised learning, the phenomenon where a predictor with good performance in some (training) domains may have poor performance when deployed in a novel (test) domain. There are two goals when faced with domain shifts. First, we would like to learn a fixed predictor that is *domain-invariant* in the sense that it has good performance in all domains. Note, however, that even a good domain-invariant predictor may still be far from optimal in any given target domain. In such cases, we'd like to learn an optimal domain-specific predictor as quickly as possible. Then, the second goal is to learn a representation for our data that is *transportable* in the sense that, when given data from a new domain, we can use the representation to learn a domain-specific predictor using only a small number of examples.

Domain shifts plague real-world applications of machine learning and there is a large and active literature aimed at mitigating the problem [e.g., Arj+19; Vei+21; PBM16; Rot+21; Wan+21; Zho+21; Koh+21; Zhu+20; Cai+21; Shi+21; Sag+19; Bai+20; SSS19; Zhe+21; Liu+21; Lu+21]. Empirically, when domain shift methods are applied to wide-ranging benchmarks, there is no single dominant method—indeed, it's common for methods that work well in one context to do worse than naive empirical risk minimization (i.e., ignore the shift problem) in another context [Koh+21; GL21]. This problem is fundamental: it is impossible to build predictors that are robust to all possible kinds of

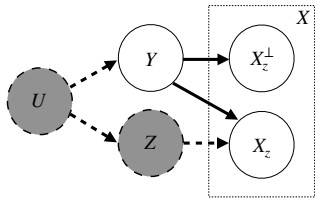

**Figure 1:** Causal model for the data generating process. We decompose the observed covariate X into latent parts defined by their causal relationships with Z. Solid circles denote observed variables, while shaded circles denote hidden variables.

shifts.[1] Accordingly, it is necessary to specify the manner in which the training and test domains are related to each other; that is, what structure is common to all domains, and what structure can vary across them. Then, to make progress on the domain shift problem, the task is to identify structural assumptions that are well matched to real-world problems and then find methods that can achieve domain-invariance and transportability under this structure.

In this paper, we rely on a particular variant of the assumption that structural causal relationships are invariant across domains, but certain "non-causal" relationships may vary. The motivation is that relationships fixed by the underlying dynamics of a system are the same regardless of the domain [PJS17]. A similar causal domain-structure assumption is already well-studied in the domain-shift literature [e.g., PBM16; Arj+19; Rot+21; Roj+18; MBS13]. There, the goal is to predict the target label $Y$ using only its causal parents (reconstructed from observed features $X$). In particular, the aim is to build predictors that do not rely on any part of the features that is causally affected by $Y$. However, in many problems, it can happen that the observed covariates $X$ are all caused by $Y$, so that the causal parents of $Y$ are the empty set. In this case, the naive causally invariant predictor is vacuous.

The purpose of this paper is to study an alternative causal notion of domain shift that handles such "anti-causal" ($Y$ causes $X$) problems, and that maintains the interpretation that structural causal relationships are held fixed across all domains. Specifically,

1. We formalize the anti-causal domain shift assumption.

2. We show how the causal domain shift assumption can be leveraged to find an invariant predictor and transportable representation.

3. We use this as the basis of a concrete learning procedure for domain-invariant and domain-adaptive representations.

## 2   Causal Setup

The first step is to make precise what structure is preserved across domains, and what structure varies. Once we have this, we'll make the notions of invariant and fast-adapting predictor precise.

In each domain $e$, we have observed data $(X_i, Y_i) \overset{\text{iid}}{\sim} P^e$, where $P^e$ is a domain-specific data-generating distribution. We will mainly consider problems where we have (finite) datasets sampled from multiple distinct domains at training time, and wish to make predictions on data sampled from some additional domains not observed during training. This paper discusses classification problems where $Y_i$ is discrete.

To formalize the causal structure assumption, we'll introduce two additional latent (unobserved) variables. First, $Z$, a (subset of) the causes of $X$. Second, $U$, a confounding variable that affects both $Z$ and $Y$; see Figure 1. Conceptually, $Z$ are the factors of variation where the association with $Y$ can vary across domains. The confounder $U$ is the reason the association can vary. The relationship between $Y$ and $Z$ induced by $U$ needs not be stable across domains. Slightly abusing notation, we have that $Z_i, U_i, X_i, Y_i \overset{\text{iid}}{\sim} P^e$ in each domain $e$. We can now describe the set of domain shifts we consider.

---

[1]For any given predictor, it's possible to adversarially construct a domain where that predictor does poorly.

**Definition 1.** (Compatible Anti-Causal Shift Domains) Distributions $\{P^e\}$ (over $X, Y$) are *compatible anti-causal shift domains* if the following conditions hold. First, there are unobserved variables $Z, U$ such that causal graph in Figure 1 holds in all domains. Second, there is a fixed distribution $P$ and for each $e$ there is some distribution $Q^e(U)$ such that $P^e(X, Y, Z) = \int P(X, Y, Z \mid U) \mathrm{d}Q^e(U)$.

Informally: The causal structure is fixed in all domains (implying the conditional distribution over X, Y, Z given U is the same). We allow *only* the distribution of the unobserved common cause $U$ to vary.

This notion of domain shift respects the preserved-causal-structure desiderata. However, it is not obvious that it suggests any useful algorithms for learning robust predictors. This is the subject of the remainder of the paper.

## 2.1 Invariant Prediction

Intuitively, a predictor will be robust against domain shifts if it depends only on causes of $X$ that have a stable relationship with $Y$ in all domains. In our setup, these are the factors of variation that are not included in $Z$. Accordingly, we want a predictor that depends only on the parts of $X$ that are not causally influenced by $Z$.

To formalize this notion, we'll use the concept of *counterfactual invariance* to $Z$ [Vei+21]. A function $f$ is counterfactually invariant to $Z$ if $f(X(z)) = f(X(z'))$ for all $z, z'$, where $X(z)$ denotes the counterfactual $X$ we would see had $Z$ been $z$. Learning a predictor that does not depend on the factors of variation $Z$ that induce unstable relationships means learning a predictor that is counterfactually invariant to $Z$.

Part of our goal in the following will be to learn a counterfactually invariant predictor. This causal notion of invariance is closely related to the notion of invariance that requires a predictor to be the risk minimizer in all domains [PBM16; Arj+19]. Specifically, under the anti-causal structure, if prior distributions $P^e(Y)$ are the same in all domains, then Veitch et al. [Vei+21, Thm. 4.2] shows that if $f$ is the counterfactually invariant predictor with the lowest risk in any training domain, it is also the counterfactually invariant predictor with the lowest risk in all domains. Even when $P^e(Y)$ is not constant across domains—there's a prior shift—imposing counterfactual invariance should still improve out-of-domain performance since it removes the domain-varying part of the features $X$. This is supported by experiments in Appendix B.2. Throughout the paper, we use the term "invariant" in the sense of counterfactual invariance.

## 2.2 Causal Decomposition of $X$

To go further in our formalization, we'll need another idea from Veitch et al. [Vei+21]: the decomposition of $X$ into (latent) parts defined by their causal relationship with $Z$. We define $X_z^\perp$ to be the part of $X$ that is not causally affected by $Z$. More precisely, $X_z^\perp$ is the part of $X$ such that any function of $X$ is counterfactually invariant if and only if it is a function of $X_z^\perp$ alone (that is, $f(X)$ is $X_z^\perp$ measurable). Under weak conditions on $Z$, $X_z^\perp$ is well defined (e.g., discrete $Z$ suffices) [Vei+21].

We also introduce $X_z$ to denote the part of $X$ that is not invariant to $Z$. We make the extra assumption that $X_z^\perp$ does not have a causal effect on $X_z$ (the other direction is ruled out by the definition of $X_z^\perp$). The meaning of this assumption is that the $Z$-specific parts of $X$ can be disentangled in the sense that it's possible to vary the other parts of $X$ without affecting $X_z$. For example, we can change the object of the image without changing the background. This is a non-trivial assumption about the structure of the anti-causal shift domains, baked into the causal compatibility assumption by the absence of an arrow between $X_z$ and $X_z^\perp$.

## 2.3 Rapid Adaptation

Even if $P(Y)$ is held fixed, the optimal counterfactually invariant predictor $g(X)$ is unlikely to be the best predictor in any given domain. The reason is that it excludes $Z$-dependent information that may in fact be highly predictive in a given domain. Given a new domain $e$, we would like to be able to quickly learn a new predictor $f^e(X)$ that updates the invariant predictor with domain-specific associations. This update should only depend on $X_z$ because the relation between $X_z^\perp$ and $Y$ is stable. Accordingly, we want to learn a representation $h(X)$ that encapsulates the information in

$X_z$. Moreover, this should be done in a manner such that, given $g(X)$ and $h(X)$, we can learn a good predictor for $P^e$ with only a small number of samples.

To formalize this, we'll introduce the following domain-specific predictors:

$$f^e(X) = g(X) + M^e h(X) \tag{2.1}$$

Here $f^e(X)$ is logits. In words: $f^e$ adds a correction to the invariant predictor $g$ that is logit-linear in the learned representation $h(X)$. We take the correction to be a linear map because, once $h$ is known, linear maps are very sample efficient to learn. Accordingly, we can formalize "learn $h$ such that we can rapidly adapt in new domains" as "learn $h$ such that the domain-specific predictor with optimal $M^e$ has low risk under $P^e$". Then, our second goal is to learn such a representation $h$.

### 2.4  Learning Goals

We have now given a causal formalization of the domain transfer scenario we consider, and formalizations of the problems of learning invariant and rapidly adapting predictors. With the causal notation in hand, our goal can be plainly stated. We want to learn an invariant $g(X)$ and a domain-varying $h(X)$ with the following properties.

1. $g(X)$ depends only on $X_z^\perp$.
2. $g(X)$ has low risk in each training domain.
3. $h(X)$ depends only on $X_z$.
4. $f^e(X) = g(X) + M^e h(X)$ should have low risk in each training domain, where $M^e$ is the linear map that minimizes the domain-specific risk.

The challenge now is that we do not observe $Z$ (or $U$) for any data point and we do not know the decomposition of $X$ into $X_z^\perp$ and $X_z$. As we will see in the next section, we can find a relaxation that is enforced with observed data which relies on the particular anti-causal structure.

## 3  Observable Signature

The first problem we must confront is how to learn a function $g(X)$ that depends on $X_z^\perp$ alone, and $h(X)$ that depends on $X_z$ alone. Strictly speaking, learning such functions precisely would be impossible, even if we observed $Z$ [Vei+21]. The reason is that we have access to only observational data, but the two parts of $X$ are defined in terms of the underlying causal structure. Instead, the best we can hope for is to require that $g(X)$ and $h(X)$ satisfy the observable implications of the causal structure. That is, the properties of the causal assumption that can actually be measured using the observed data.

When $Z$ is observed, a signature for $g(X)$ is that $g(X)$ is conditionally independent of $Z$ given $Y$ [Vei+21]. But when $Z$ is unobserved, it is challenging to learn $g(X)$ and $h(X)$, as these representations of $X$ are intimately tied to $Z$.

The key observation is that there are two relations that connect $g(X)$ and $h(X)$. The first relation comes from the causal graph. In particular, we want to impose the requirement that $g(X)$ and $h(X)$ satisfy the observable implications of the causal structure. The next theorem gives such an observable implication, which can serve as an observable signature of the causal decomposition.

**Theorem 2.** *If $g(X)$ depends only on $X_z^\perp$ and $h(X)$ depends only on $X_z$, then, under the causal graph in Figure 1, $g(X) \perp\!\!\!\perp h(X) \mid Y$.*

The usefulness of this theorem is that the conditional independence statement can be measured from data, and enforced in the model training.

The second relation is subtler and it comes from our formulation of domain-specific predictors $f^e$. Specifically, $f^e$, as a linear combination of $g$ and $h$, should minimize the risk in every domain. And by only allowing coefficients of $h$ to change, we hope $g$ would capture "invariant" information ($X_z^\perp$) and $h$ would learn "unstable" information ($X_z$).

Therefore, to try to enforce conditions 1 and 3 in Section 2.4, we can learn $g$ and $h$ jointly to minimize domain-specific risk, while enforcing that $g(X)$ and $h(X)$ satisfy the conditional independence

implied by the causal structure. Since the observable signature is only necessary (not sufficient) for the causal decomposition and there could be multiple candidates of $g$ and $h$ pairs that can parameterize $f^e$ in the aforementioned way, it's not guaranteed to recover $g$ and $h$ that only rely on $X_z^\perp$ and $X_z$ respectively. However, it does strongly constrain the functions we can learn. And, as we will see in Section 6, enforcing the signature does lead to predictors with good robustness and fast adaptation properties.

## 3.1 Causal Regularization

We enforce $g(X)$ and $h(X)$ to satisfy the conditional independence condition via regularization. Specifically, we want to define a regularizer such that its value goes to zero whenever the conditional independence requirement is met. In general, measuring conditional independence is hard [Zha+12; Fuk+07; TSS16]. Instead, we enforce a weaker condition that uses the following fact.

**Lemma 3.** *If* $A \perp\!\!\!\perp B \mid D$, *then,* $\mathbb{E}[A \cdot (B - \mathbb{E}[B|D])] = 0$

This is a necessary but not sufficient condition for conditional independence. However, it is easy to compute and leads to good results in practice (as shown in Section 6). With this identity in hand, we define $C_{\text{cond}}(A, B, D)$, the (infinite data) conditional independent regularization term between random variables $A, B$ given random variable $D$, and its empirical estimate $\hat{C}_{\text{cond}}$ as follows:

$$C_{\text{cond}}(A, B, D) = \mathbb{E}[A \cdot (B - \mathbb{E}[B|D])]$$

$$\hat{C}_{\text{cond}}(\{(a_i, b_i, d_i)\}_{i=1}^n) = \left\| \frac{1}{n} \sum_i a_i \left( b_i - \frac{1}{|\#j : d_j = d_i|} \sum_{j:d_j=d_i} b_i \right) \right\|_1$$

where $\{a_i\}_{i=1}^n$, $\{b_i\}_{i=1}^n$, $\{d_i\}_{i=1}^n$ are samples of $A, B$ and $D$. Here, the conditional random variable $D$ is assumed to be discrete which is true for the use case in this paper.

# 4 Learning Algorithm

We have reduced our goal to learning $g$ and $h$ such that

1. For training domains, $g(X)$ has low risk.
2. For a given domain $e$, there exists $M^e$ such that $f^e(X) = g(X) + M^e h(X)$ is the risk minimizer of that domain.
3. $g(X), h(X)$ are constrained by the conditional independence regularization.

We now design a specific algorithm that accomplishes the learning task. First, we parameterize the learning problem in a form that's convenient to use with neural networks. Then, we translate our learning objectives into a bi-level optimization problem. Finally, we introduce a practical algorithm to solve the bi-level optimization problem. We name our method ACTIR for **A**nti- **C**ausal **T**ranportable and **I**nvariant **R**epresentation.

## 4.1 Reparameterization

In principle, we could learn two completely separate functions $g(X)$ and $h(X)$. However, this can be wasteful. For instance, in vision, many low-level features can be reused by different predictors. To address this, we first notice that we can always rewrite (2.1) as follows

$$f^e(X) = (W^b + W^e)\Phi(X). \tag{4.1}$$

That is, as a shared representation $\Phi$ followed by a fixed linear map $W^b$ defining $g$ and a domain-specific linear map $W^e$ defining $M^e h(X)$.[2] The task is then learning the representation (which we'll parameterize by a neural network), and the invariant and domain-specific linear maps.

In fact, a further simplification is possible: we can fix $W^b$ to be $\begin{bmatrix} \mathbf{I} & \mathbf{0} \\ \mathbf{0} & \mathbf{0} \end{bmatrix}$. The reason is that, because $\Phi$ is unconstrained, learning $W^b$ doesn't actually add expressive power—any non-zero map suffices.

---

[2]Consider $\Phi(X) = [g(X)^T, h(X)^T]^T$, $W^b = \begin{bmatrix} \mathbf{I} & \mathbf{0} \\ \mathbf{0} & \mathbf{0} \end{bmatrix}$ and $W^e = \begin{bmatrix} \mathbf{0} & \mathbf{0} \\ \mathbf{0} & M^e \end{bmatrix}$.

## 4.2 Bi-Level Optimization

We have now reduced our task to a bi-level optimization problem

$$
\begin{aligned}
\min_{\Phi} \sum_{e \in \mathcal{E}_{tr}} & \gamma R^e((W^b + W^e)\Phi) + (1 - \gamma)R^e(W^b\Phi) \\
& \text{st } W^e \in \underset{W}{\arg\min}\, R^e((W^b + W)\Phi) + \lambda C_{\text{cond}}(W^b\Phi, W\Phi, Y) \quad \forall e \in \mathcal{E}_{tr}
\end{aligned}
\tag{4.2}
$$

where $\gamma \in [0, 1]$, $\lambda > 0$. The set $\mathcal{E}_{tr}$ consists of all training domains, and $R^e(f)$ is the domain-specific population risk defined as $R^e(f) = \mathbb{E}_{(\mathbf{x},\mathbf{y}) \sim P^e}[\ell(f(\mathbf{x}), \mathbf{y})]$ with the cross-entropy loss function $\ell$.

In words: we try to learn a representation $\Phi$ such that the invariant predictor has low risk (second term), the domain-specific predictor has low risk in each domain (first term), and the domain-specific perturbation $W^e$ is optimal given $W^b$ and $\Phi$ while satisfying the observable signature of the causal condition (constraint, with the $C_{\text{cond}}$ regularization).

## 4.3 Practical Algorithm

(4.2) is a challenging optimization problem. In general, each constraint calls for an inner optimization routine. So instead of solving (4.2) directly, we use a gradient penalty to make the problem more tractable. Specifically, we translate the condition that the domain-specific risk is optimal (the inner loop) into the condition that the gradient of the domain-specific risk with respect to $W^e$ is 0. Then, we regularize the $\ell_2$-norm of this gradient. This is inspired by a similar trick used in Invariant Risk Minimization [Arj+19]. The finite sample objective function can be expressed as:

$$
L(W^b, W^e, \Phi) =
$$

$$
\sum_{e \in \mathcal{E}_{tr}} \left[ \sum_{(x,y) \in D^e} \gamma\ell\big((W^b + W^e)\Phi(x), y\big) + (1-\gamma)\ell\big(W^b\Phi(x), y\big) \right] + \lambda_g \sum_{e \in \mathcal{E}_{tr}} \|\nabla L^e_{\text{inner}}(W^e)\|^2
$$

where $\lambda_g > 0$ is a regularization coefficient for the gradient penalty, $D^e$ is a labeled dataset collected from training domain $P^e$ and $L^e_{\text{inner}}$ is given by

$$
L^e_{\text{inner}}(W) = \sum_{(x,y) \in D^e} \ell\left((W^b + W)\Phi(x), y\right) + \lambda\hat{C}_{\text{cond}}\left(\left\{(W^b)\Phi(x), W\Phi(x), y\right\}_{(x,y) \in D^e}\right)
$$

## 4.4 Invariant and Adaptive Prediction

After training, suppose the returned representation is $\hat{\Phi}$. Then the invariant predictor is

$$
g(X) = W^b\hat{\Phi}(X)
$$

Moreover, given a few labeled examples from a new domain, we can find a domain-specific predictor by fine-tuning the linear layer $W^b$.

# 5 Related Work

**Causal Prediction**  Several papers connect causality and robustness to domain shifts. [e.g., PBM16; HDPM18; Arj+19; Lu+21]. These papers usually assume that all domains share a common causal structure, and consider the set of domains induced by arbitrary intervention on any node other than the label $Y$. In this case, the predictor that has invariant risk across domains is the one that depends only on the causal parents of $Y$. By contrast, in this paper, we only allow changes of unobserved confounders—resulting in a much smaller set of possible shifts. Restricting the possible shifts enlarged the set of possible invariant predictors, allowing for invariant predictors that depend on the descendants of $Y$.

A closely related work is Invariant Risk Minimization [Arj+19], that also seeks to learn a representation $\Phi$ of $X$ such that a fixed linear map on top of the representation yields an invariant predictor. The major distinction with the approach here is that we have a different notion of invariance (see

Section 2.1), and we rely on simultaneously learning the non-stable factors of variation in order to identify the representation.

Other papers also consider settings where the covariates $X$ are not direct causes of $Y$ [Liu+21; Mit+21; Ils+20]. They assume that both $X$ and $Y$ are caused by latent variables that can be divided into stable and non-stable parts. Then, they use generative models reflecting this assumption. By contrast, the approach in this paper is fully nonparametric—there is no explicit modeling of the generative process. Prediction in the anti-causal direction has also been studied in other contexts [Sch+12; Li+18; Wal+21; KPS18; Gon+16; HM21]. In particular, Schölkopf et al. [Sch+12] study the role of anti-causal learning in semi-supervised learning and transfer learning.

This work fits into the emerging literature on causal representation learning [e.g., Bes+18; Loc+20; Sch+21; WJ21]. This literature seeks to find representations that disentangle causally meaningful components of the data—here, we disentangle the factors of variation that have domain-stable or domain-varying relationships with the target $Y$.

Veitch et al. [Vei+21] introduce the notion of counterfactual invariance to a spurious factor and make some connections with domain shifts. However, they assume $Z$ is known in advance and observed, and rely on this to learn the counterfactually-invariant predictor. In contrast, in this paper we merely assume the existence of some $Z$—we don't need to know it in advance, and we don't need to measure it directly. And, they use data from only a single domain, whereas we require observations from several distinct domains. We also treat the problem of learning transportable representations, but they only handle invariant learning.

**Domain Adaptation and Meta Learning** There have been numerous fruitful developments in the fields of domain generalization and adaptation [e.g., Zho+21; Wan+21], including ones under various causal assumptions [Zha+13; Mag+18; CB21; SSS19; Sch+12; Lv+22]. A distinctive aspect of the work in this paper is that we consider the interplay between both the problem of invariant/robust learning and adaptation.

The adaptive part of the learning model in this paper is also related to meta learning, where the goal is to learn predictors that can quickly adapt to new tasks. Meta learning has been used for supervised learning [San+16], reinforcement learning [Wan+16] and even unsupervised learning [JV19]. Traditional approaches to meta learning include defining a distribution over the structure of input data to perform inference [Lak+11] or to use a memory model such as LSTM [Wan+18]. But the dominant models for meta learning are generic gradient-based learning methods such as MAML [FAL17] and Reptile [NAS18]. Theoretically, Tripuraneni et al. [TJJ21] and Du et al. [Du+20] also examine the representation power of meta-learning. Although not motivated by causality, they show that if there is a shared common structure, meta learning can be used to reduce sample complexity in unseen domains.

## 6 Experiments

The main claims of the paper are:

1. The invariant predictor $g(X)$ will have good performance in new domains, so long as the shifts obey the anti-causal structure.

2. The learned representation $\Phi$ enables rapid adaptation to new domains by learning only a linear adjustment term on top of $\Phi$.

3. The learned $\Phi$ disentangles the parts of $X$ that are not affected by $Z$ from the parts that are.

To evaluate the above claims, we conduct experiments on synthetic and real-world data. While causal structures of real-world problems like image classification are usually unknown, we find that the anti-causal based method works well on many such problems—suggesting the anti-causal structure is appropriate. For space reasons, in addition to the experiments reported in this section, we also report experiments on PACS [Li+17] and VLCS [FXR13] in Appendix B. These experiments also strongly support the effectiveness of the causal adaptive model. We also provide a counterexample showing that ACTIR can fail when causal assumptions fail to hold in, Appendix B.3.

**Baselines** For each experiment, all methods share a common architecture; they differ only in objective functions or optimization procedures. For invariant learning, we compare with empirical risk minimization (ERM), IRM [Arj+19] and the MAML [FAL17] base learner. To test how well learned

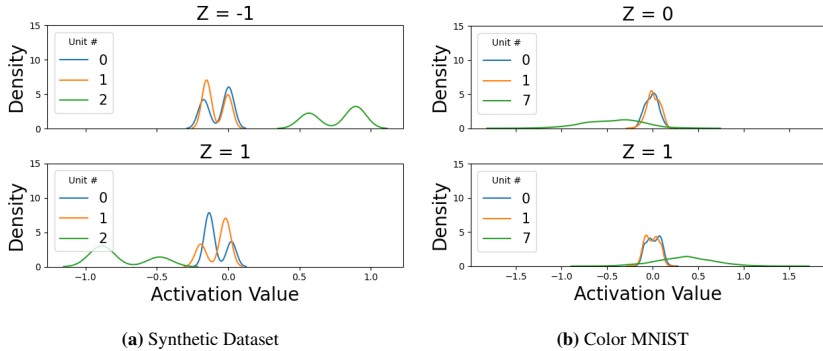

**(a)** Synthetic Dataset        **(b)** Color MNIST

**Figure 2:** The learned representation $\Phi$ disentangles stable and unstable factors of variation. Plots show activation levels of different representation units. Units 0 and 1 are trained as the invariant part of the representation. The invariant units have no dependency on the unstable factor $Z$, but the other unit has a strong dependency. For synthetic dataset, $Z$ is the random variable defined in the structural equations. For Color MNIST, $Z$ is the color.

representations $\Phi$ can enable fast adaptation, we fine-tune linear models on top of the representation. For comparison, we also fine-tune linear layers on top of the representations (penultimate layers) from ERM, IRM, and MAML. It has been shown recently that fine-tuning the last layer of models trained by ERM has surprisingly good performance on many real-world datasets [RRR22]. For MAML, the last layer is trained using the MAML update rule.

## 6.1 Synthetic Dataset

We generate synthetic data according to the following structural equations (which obey the anti-causal structure):

$$Y \leftarrow \mathrm{Rad}(0.5) \quad X_z^\perp \leftarrow Y \cdot \mathrm{Rad}(0.75) \quad Z \leftarrow Y \cdot \mathrm{Rad}(\beta_e) \quad X_z \leftarrow Z$$

where input $X$ is $(X_z, X_z^\perp)$ and $\mathrm{Rad}(\beta)$ means that a random variable is $-1$ with probability $1 - \beta$ and $+1$ with probability $\beta$. We create two training domains with $\beta_e \in \{0.95, 0.7\}$, one validation domain with $\beta_e = 0.6$ and one test domain with $\beta_e = 0.1$. Prediction with $X_z^\perp$ is stable but has a lower accuracy compared to prediction with $X_z$ during training. If a learning model only chooses the classifier with the best prediction accuracy in training domains and ignores its instability, it will choose $X_z$ as its predictor and end up with only $10\%$ accuracy on the test set. The robust predictor would be $X_z^\perp$ with $75\%$ accuracy. On the other hand, in the test domain, $-X_z$ predicts $Y$ with $90\%$ accuracy—so an adaptive predictor is better than the invariant one.

We use a three-layer neural network with hidden size $8$ and ReLU activation for $\Phi$ and train the neural network with Adam optimizer. The hyperparameters are chosen based on performance on the validation set. For the fine-tuning test, we run 20 steps with a learning rate $10^{-2}$. The result is shown in Table 1. Both IRM and ACTIR learn good invariant predictors. But ACTIR is also equipped with the ability to adapt given a very small amount of data points while the performance of IRM stays the same after fine-tuning. Perhaps unsurprisingly, ERM has a test accuracy of $10\%$, suggesting that it uses only spurious features $X_z$.

## 6.2 Color MNIST

Color MNIST modifies the original MNIST dataset [Arj+19]. First, we assign label 0 to digits 0-4 and label 1 to digits 5-9. We then flip the label with probability $25\%$ and assign colors to the original images based on the label but with a flip rate $1 - \beta^e$. That is, we assign color 0 to images with label 0 with probability $\beta^e$. Here, color is naturally the factor of variation $Z$. We create two training domains with $\beta^e \in \{0.95, 0.7\}$, a validation domain with $\beta^e = 0.2$ and a test domain with $\beta^e = 0.1$. Ideally, we want to learn invariant predictors based on the shape of the digit—this will achieve $75\%$ accuracy. But the problem is significantly more challenging than the synthetic example because the color is a much easier feature to learn than the shape of the digit, making models more susceptible to spurious correlations. We use a three-layer convolutional neural network for $\Phi$ and train the neural network with Adam optimizer. The hyperparameters are chosen based on performance on the validation set.

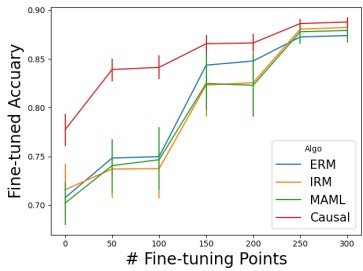

**Figure 3:** ACTIR has good accuracy after fine-tuning with small datasets on Camelyon17. Graph shows accuracy (%) on Camelyon17 for different numbers of fine-tuning examples. Standard errors are over 5 runs. For each run, the accuracy is averaged for 100 fine-tuning tests.

**Table 1:** ACTIR has good invariant and adaptive performance on synthetic datasets and Color MNIST, and it outperforms baseline methods on Camelyon17 for invariant prediction. The table shows accuracy (%). Note that adaptation ($n$) means that the predictor is tuned on $n$ number of labeled examples. For synthetic datasets, standard errors are over 100 runs. For Color MNIST, it is over 50 runs. For Camelyon17, it is over 5 runs. In each run, the adaptive accuracy is determined by the average of 100 fine-tuning tests for both synthetic datasets and Color MNIST.

| | SYNTHETIC DATASET | | COLOR MNIST | | CAMELYON17 |
| METHOD | TEST ACC. | ADAPTATION (10) | TEST ACC. | ADAPTATION (10) | TEST ACC. |
| --- | --- | --- | --- | --- | --- |
| ERM | 9.95±0.10 | 11.57±0.71 | 28.24±0.51 | 27.26±0.48 | 70.77±1.98 |
| IRM | 74.91±0.13 | 74.27±0.47 | 59.97±0.91 | 60.16±0.90 | 71.59±2.76 |
| MAML | 17.14±2.22 | 44.01±3.48 | 22.18±1.01 | 75.03±3.30 | 70.22±2.40 |
| **ACTIR** | 74.77±0.44 | **89.28±0.25** | **70.30±0.71** | **85.25±1.11** | **77.73±1.74** |

For the fine-tuning test, we run 20 steps with a learning rate $10^{-2}$. The result is shown in Table 1. ACTIR learns both the invariant and adaptive structure significantly better than reference baselines.

**Causal Disentanglement** To understand why ACTIR can adapt to test domains in both synthetic and Color MNIST datasets, we plot distributions of activation values of $\Phi$. See Figure 2. We see that the first two coordinates—used as the invariant part of the representation in training (see Section 4.3)—have distributions that do not depend on the value of $Z$. On the other hand, some other (non-invariant) representation coordinates have activations that change dramatically depending on the value of $Z$. Thus, the representation effectively disentangles the $X_z^\perp$ and $X_z$ features. Importantly, this is achieved with no a priori knowledge of what $Z$ might be, and no observations of it.

### 6.3 Camelyon17

The goal of the Camelyon17 dataset [Ban+18] is to predict the existence of a tumor given a region of tissue. This is a binary classification problem. Data are collected from a small number of hospitals. But there are variations in data collection and processing that could negatively impact model performance on data from a new hospital. We take the individual hospitals to be separate domains. The objective is to generalize to new hospitals not seen in training. The dataset consists of input images with size $96 \times 96$ and binary labels that indicate if the central $32 \times 32$ regions contain any tumor tissues. The dataset can be divided into 5 subsets, each from a different hospital. Following the WILDS benchmark [Koh+21], we use 3 for training, 1 for validation, and the last one for test.

We use a pre-trained ResNet-18 model for our $\Phi$ and train the whole model using Adam optimizer with a learning rate $10^{-4}$. For the fine-tuning test, we run 20 iterations with a learning rate $10^{-2}$. As shown in Table 1, ACTIR has the best invariant accuracy. For adaptive performance, Figure 3 shows that ACTIR has a large performance improvement when given a small fine-tuning dataset, while other models require more fine-tuning examples to see a significant increase in accuracy.

# 7 Discussion

This paper studies learning invariant and transportable representations for a specific class of anti-causal shift domains. We assume that all domains have a common anti-causal structure and are differentiated only by the distribution of certain unobserved confounders. This setup is a reasonable match for many practical problems.

This work serves as a proof of concept for this anti-causal domain shift notion, showing that it can be translated into useful learning principles for domain adaptation. This opens the door for substantial future work. In particular, the practical training procedure we use can likely be refined. It would also be nice to find formal guarantees for robust models trained under this setup—e.g., relying on some notion of diversity of training domains.

# 8 Acknowledgments

Thanks to Jacob Eisenstein for feedback on an earlier version. We also acknowledge the University of Chicago's Research Computing Center for providing computing resources. This work was partially supported by Open Philanthropy.

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
