# A  Proofs

**Theorem 2.** *If $g(X)$ depends only on $X_z^\perp$ and $h(X)$ depends only on $X_z$, then, under the causal graph in Figure 1, $g(X) \perp\!\!\!\perp h(X) \mid Y$.*

*Proof.* Reading d-separation from the causal graphs. We have that $X_z \perp\!\!\!\perp X_z^\perp \mid Y$. $g(\cdot)$ is a function of $X_z^\perp$ and $h(\cdot)$ is a function of $X_z$. Because function of independent variables are also independent, the theorem is proven. $\qquad\square$

**Lemma 3.** *If $A \perp\!\!\!\perp B \mid D$, then, $\mathbb{E}[A \cdot (B - \mathbb{E}[B|D])] = 0$*

*Proof.*

$$\begin{aligned}
\mathbb{E}[A \cdot (B - \mathbb{E}[B|D])] &= \mathbb{E}[\mathbb{E}[A \cdot (B - \mathbb{E}[B|D])]|D] \quad \text{law of total expectation} \\
&= \mathbb{E}[\mathbb{E}[A|D] \cdot \mathbb{E}[(B - \mathbb{E}[B|D])|D] \quad \text{conditional independence} \\
&= 0
\end{aligned}$$

$\square$

# B  Additional Experiments

## B.1  PACS

PACS [Li+17] consists of four domains: art painting, cartoon, photo, and sketch. Each image is of size $\{3, 224, 224\}$ and belongs to one of 7 classes: dog, elephant, giraffe, guitar, horse, house, person. The style of the image could be confounded with the label, which is why an invariant predictor is desirable.

For each domain, we test the model performance by training on the other three domains. The hyperparameters are chosen using leave-one-domain-out cross-validation [GL21]. This means that we train 3 models by leaving one of the training domains out and using it as a validation set. We choose the hyperparameters that have the highest average performance of these 3 models on validation sets. And then we retrain the model with all training domains using the newly selected hyperparameters. To test adaptivity, we fine-tune the last layer of the model by randomly selecting examples in the target domain.

We re-sample datasets such that label distributions are balanced and consistent across all domains. We use a pre-trained ResNet-18 model for our feature extractor and train the whole model using Adam optimizer with a learning rate $10^{-4}$. For the fine-tuning test, we run 20 steps using Adam with a learning rate $10^{-2}$. Table 2 shows that ACTIR has competitive invariant performance compared to baselines. Furthermore, Figure 4 demonstrates that ACTIR can also adapt given a small fine-tuning dataset. ACTIR does not perform very well on the Sketch domain (S) for both invariant and adaptive predictions. In this case, all training domains: art, cartoon, and photo have colors while the test domain does not. We suspect training domains are just not diverse enough for the model to successfully disentangle invariant and adaptive features.

**Table 2:** ACTIR has good invariant performance on PACS dataset. Table shows accuracy (%) on PACS. Standard errors are over 5 runs.

| METHOD | A | C | P | S |
|---|---|---|---|---|
| ERM | 79.30±0.50 | 74.30±0.73 | 93.03±0.26 | 65.40±1.49 |
| IRM | 78.69±0.70 | 75.38±1.49 | 93.32±0.32 | 65.61±2.55 |
| MAML | 74.18±4.00 | 75.15±1.66 | 91.05±1.09 | 63.56±3.88 |
| **ACTIR** | **82.55±0.45** | **76.62±0.65** | **94.17±0.12** | 62.14±1.30 |

## B.2  VLCS

For all previous experiments, the prior distributions $P(Y)$ are the same for all domains. As suggested in Section 2.1, if the prior distributions are the same, then the best counterfactually-invariant predictor

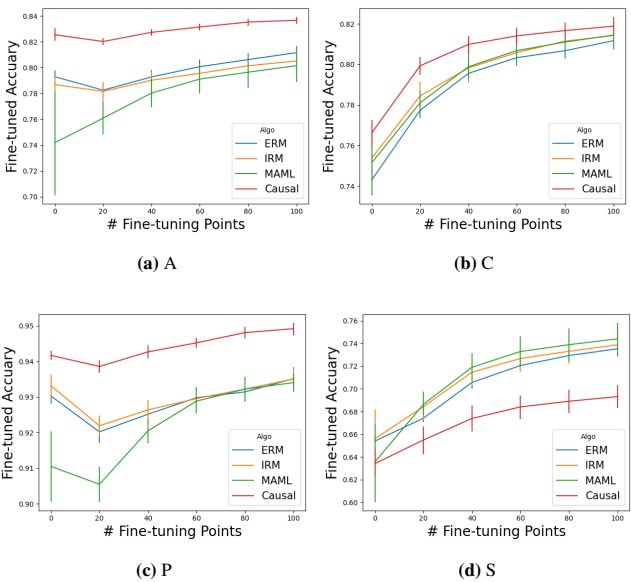

**(a)** A  **(b)** C  **(c)** P  **(d)** S

**Figure 4:** ACTIR has the best accuracy after fine-tuning for all but one domain. Graph shows accuracy (%) on PACS. Standard errors are over 5 runs. For each run, the accuracy is averaged for 100 fine-tuning tests.

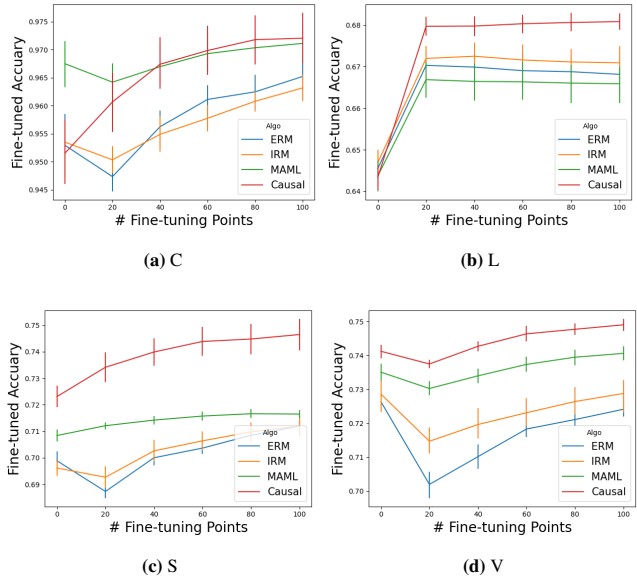

**(a)** C  **(b)** L  **(c)** S  **(d)** V

**Figure 5:** ACTIR has competitive performance after fine-tuning on all domains. Graph shows accuracy (%) on VLCS. Standard errors are over 5 runs. For each run, the accuracy is averaged for 100 fine-tuning tests.

in the training domain will also be the best counterfactually-invariant predictor in the test domain. But requiring label distributions to stay consistent is too stringent. In practice, however, ACTIR still works reasonably well even when classes are not balanced. We'll see this with experiments on VLCS [FXR13]. VLCS contains four photographic domains. Each image is of size $\{3, 224, 224\}$ and belongs to one of $5$ classes: bird, car, chair, dog, and person. For each domain, images are collected differently. Ideally, an invariant predictor should be indifferent to the photo collecting process.

Similar to Appendix B.1, we also use a pre-trained ResNet-18 model for our feature extractor and train the whole model using Adam optimizer with a learning rate $10^{-4}$. The training procedure follows exactly as in Appendix B.1. The hyperparameters are also chosen using leave-one-domain-out cross-validation. For the fine-tuning test, we run 20 steps with a learning rate $10^{-2}$. For MAML, the fine-tuning learning rate is set to $10^{-3}$. Otherwise, the result for MAML is unstable. Table 3 shows

that ACTIR predictor has competitive invariant performance compared to baselines. Furthermore, Figure 5 demonstrates that ACTIR can also adapt given a small fine-tuning dataset. In particular, for the LabelMe dataset (L), four models have similar accuracy before fine-tuning and ACTIR has the best accuracy after fine-tuning.

**Table 3:** ACTIR has good invariant performance on VLCS dataset. Table shows accuracy (%) on VLCS. Standard errors are over 5 runs.

| METHOD | C | L | S | V |
|--------|---|---|---|---|
| ERM | 95.29±0.58 | 64.55±0.41 | 69.89±0.42 | 72.65±0.22 |
| IRM | 95.35±0.38 | 64.69±0.34 | 69.61±0.30 | 72.86±0.58 |
| MAML | **96.75±0.47** | 64.37±0.40 | 70.84±0.24 | 73.51±0.28 |
| **ACTIR** | 95.15±0.59 | 64.34±0.31 | **72.31±0.41** | **74.12±0.21** |

## B.3 Counterexample: When the Data Generating Process Does not Fit Causal Assumptions

**Table 4:** IRM can outperform ACTIR on data not obeying the anti-causal structure. Table shows accuracy (%) on synthetic dataset. Note that adaptation ($n$) means that the predictor is tuned on $n$ number of labelled examples. Standard errors are over 100 randomly generated datasets, for both initial training and adaptation.

| METHOD | TEST ACC. | ADAPTATION (5) | ADAPTATION (10) |
|--------|-----------|----------------|-----------------|
| ERM | 11.57±0.71 | 11.58±0.71 | 11.59±0.71 |
| IRM | **69.61±1.26** | **69.61±1.26** | **69.61±1.26** |
| MAML | 11.57±0.71 | 11.83±0.75 | 11.93±0.78 |
| **ACTIR** | 43.51±2.63 | 62.37±2.44 | 64.17±2.63 |

ACTIR assumes the anti-causal structure. Specifically, it needs the conditional independence condition implied by Theorem 2 to hold. If this condition fails, the algorithm can fail. To see this, we create synthetic datasets with the following structural equations:

$$X_y \leftarrow \text{Bern}(0.5)$$
$$Y \leftarrow \text{XOR}(X_y, U) \quad U \leftarrow \text{Bern}(0.75)$$
$$Z \leftarrow \text{XOR}(Y, U_z) \quad U_z \leftarrow \text{Bern}(\beta_e)$$
$$X_z \leftarrow \text{XOR}(Z, X_y)$$

where input $X$ is $(X_y, X_z)$ and $\text{Bern}(\beta)$ means that a random variable is 1 with probability $\beta$ and 0 with probability $1 - \beta$. We create two training domains with $\beta_e \in \{0.95, 0.8\}$, one validation domain with $\beta_e = 0.2$ and one test domain with $\beta_e = 0.1$. Here, the conditional independence no longer holds because $X_y$ directly influences $X_z$.

We use a three-layer neural network with hidden size 8 and ReLU activation for $\Phi$ and train the neural network with Adam optimizer. The hyperparameters are chosen based on performance on the validation set. For the fine-tuning test, we run 20 steps with a learning rate $10^{-2}$. The result is shown in Table 4. Because the conditional independence condition fails, ACTIR no longer has competitive invariant performance against IRM, even though it still outperforms ERM. One possible solution for this problem is to use a different causal regularizer that relies on different independence conditions. Nevertheless, ACTIR still has decent adaptive performance given a small fine-tuning set. Similar to the synthetic experiment in Section 6.1, ERM has a test accuracy of 10%, suggesting that it uses only spurious features.