# OpenReview forum: "Invariant and Transportable Representations for Anti-Causal Domain Shifts"
_NeurIPS.cc/2022/Conference — NeurIPS 2022 Accept_

### Official Review · Reviewer_wAxr · 2022-07-08

**Rating:** 7
**Confidence:** 4
**Soundness:** 3 good
**Presentation:** 3 good
**Contribution:** 4 excellent

**Summary:**

In this paper, a setting is studied where data come from different domains, the features X are causal descendants of Y, and for some part of X, the relation with Y will change between domains. Based on a notion of counterfactual invariance, an optimization objective and an algorithm are then proposed to disentangle the different parts of X. This allows prediction using just the invariant part of X, but also fine-tuning the predictor on a few samples of a new domain to leverage the information in the non-invariant part of X.

**Questions:**

1. **Definition 1**: I think the crucial aspect of this definition is not that the causal graph is shared, but that $P(X,Y,Z|U)$ is shared between domains, which is now only mentioned implicitly.

2. **counterfactual invariance**: A "counterfactual" notion of invariance is both a stronger condition than you actually need, and also stronger than can be verified from your data. It is in principle possible to detect if the *distribution* of f(X) (jointly with Y?) given z remains invariant when z changes, but not whether the specific value of f(X) for a given individual is invariant in such a way. Since the term "counterfactual" (as the third level of Pearl's hierarchy) is specifically about this latter case, I feel this term is not the best choice in this context. And more specifically, I would disagree that (line 103) learning a stable predictor "means" learning a counterfactually invariant predictor.

3. **latent parts of X**: Could you clarify in the paper what you mean by "part": a subset of the coordinates of a long vector X, or any function of X (that preserves some of the data in X but discards other parts)?

4. **Causal decomposition of X**: In the second paragraph of section 2.2, I think what you want to work towards is that the two parts of X are independent given Y. If you want to phrase this in terms of the causal diagram, I think you also need to assume here that the two parts have no latent confounders between them, and I suppose selection bias could be a problem as well. I'm not sure what is meant with this extra assumption being "baked into the causal compatibility assumption" (line 123), because the two parts of X were not defined yet when Definition 1 was posed.

5. **$W^b$ and $W^e$**: In line 215, the possibility of making $W^b$ a fixed matrix is mentioned. I assume this is also being done for the experiments? Would it also be a good simplification to restrict $W^e$ to be of the form suggested in footnote 2, or would that reduce the expressive power in an undesirable way? Relatedly, I think the $W^b$ in line 237 (about fine-tuning) should read $W^e$.

Minor points:

* it looks like the display above line 196 doesn't use \lVert and \rVert

* In three displays on page 6, summation signs are ambiguous as to which terms are part of the expression being summed.

**Strengths And Weaknesses:**

The anti-causal setting studied in this paper has not been studied before to my knowledge (except under an additive noise assumption). I think its exploration of what is possible for domain adaptation with such anti-causal variables, is a significant contribution (mainly but not exclusively for enabling future work). The derivation and methods include some elements that are derived from earlier work, but for the main part, the work is novel.

The paper is very clearly written, though I do have some questions below where e.g. a definition doesn't seem to express exactly what is intended. I hope that addressing these points will further improve this paper.

---

> ### Author Response · Authors · 2022-08-01
> **Reply to reviewer**
>
> Thank you for the detailed comments and for your support. We agree that the main contribution of the paper is enabling future work studying the (relatively neglected) anti-causal domain adaptation setting.
>
> “Definition 1”
>
> From the probabilistic perspective, you’re right that the key point is that P(X,Y,Z|U) is invariant. We have changed the text immediately following the assumption to emphasize this. We prefer the causal statement of the domain shift assumption because, although it is mathematically stronger than required, it is an assumption about the physical (causal) world. The assumption ultimately needs to be assessed based on the structural understanding of an analyst.
>
> “counterfactual invariance”
>
> It is true that, mathematically, counterfactual invariance is stronger than what we require. However, we are trying to formalize assumptions about the real, physical world. It seems more natural to us to state these assumptions in intuitive causal language (in this case, with counterfactual). As you note, it is not possible to directly measure or enforce counterfactual invariance directly—we also discuss this gap between the causal ideal and the conditional independence signature in the paper.
>
> There are different notions of what a “stable” predictor means in the literature. In the anti-causal case, there is no obvious canonical choice based on risk or similar. Part of the point here is that, under the particular causal domain shift structure we consider, counterfactual invariance to the variable features is a natural formalization (though, likely not the only possible one).
>
>
> “latent parts of X”
>
> We use the word to mean representation of X—i.e., a function $\lambda$ such that $\lambda(X)$ is measurable with respect to X, but (usually) X is not measurable with respect to $\lambda(X)$.
>
>
> “Causal decomposition of X”
>
> The causal assumptions are encapsulated in the causal diagram, including the absence of latent confounders between the parts and the non-edge from X_z^\perp to X_z. You’re right that we didn’t define the parts of X in advance of the causal compatibility assumption—we have edited the definition to note that these are defined in the following text.
>
> “W^b and W^e”
>
> Yes, we fixed W^b in experiments. We do not restrict W^e to give it more expressive power. And because W^b and W^e are linearly additive terms, fine-tuning W^b would be the same as fine-tuning W^e.
>
> We have fixed the minor issues in the revision.

---

> > ### Comment · Reviewer_wAxr · 2022-08-05
> > **Thanks**
> >
> > Thank you for the thoughtful rebuttal, this makes it clear to me that the concerns I had have been addressed.
> >
> > To reflect this, I have increased my score to 7.

---

### Official Review · Reviewer_UXLv · 2022-07-11

**Rating:** 6
**Confidence:** 4
**Soundness:** 3 good
**Presentation:** 3 good
**Contribution:** 3 good

**Summary:**

The present work address generalization and transfer of learnable representations in the setting of anti-causal domain shifts. To this end, the work formalize the anti-causal domain shift assumption by adopting the notion of counterfactual invariance (Veitch et. al., 21). Based on this formulation, the work proposes a bi-level optimization based objective to learn domain-invariant and domain-adaptive predictors in a multi-distribution training setup.  Eventually, the proposed algorithm is evaluated on a synthetic task, colored-MNIST and some real-world datasets against ERM, IRM, MAML and show promising performance.

**Questions:**

- During fine-tuning, are both the invariant features and the non-stable features adapted or only the non-stable features? Did the authors also consider to finetune with different learning rates, in order to allow an even more rapid adaptation?
- Is any regularization technique employed in ERM?

**Limitations:**

Fine

**Strengths And Weaknesses:**

**Strenghts**
- Anti-Causal problem setting is well motivated and the work is well written in general
- I appreciate the formulation based on counterfactual invariance to separate stable from unstable knowledge
- Supportive visual analysis in figure 2
- Valid sets of experimental baselines where proposed approach shows promising performance

**Weaknesses / Suggestions for improvements**
- **Evaluation**: I have some concerns about the real-world applicability of the proposed algorithm and would appreciate some discussion on this matter
  - There is a significant gap between performances on manually crafted tasks (synthetic setting, color MNIST) which match the underlying assumptions of the framework and on the real-world datasets (Camelyon17, PACs, VLCS). I would like to encourage the authors to address this issue and it's implications for applications on real-world datasets in more detail in the main part (especially including the important findings on real-world datasets in the main part of the paper).
   - What conditions need the training environments to fullfill and what dataset characteristics can break the algorithm? From my point of view, it looks like diversity of the training environment is a key property?
   - Would it be possible to show a similar analysis as figure 2 for one of the real-world datasets?
- **Assumptions**
   - The work assumes that $X_{z}^{\perp}$ does not have a causal effect on $X_z$. What are the implications on potential application(especially on structured datasets)?

---

> ### Author Response · Authors · 2022-08-01
> **Reply to reviewer**
>
> Rebuttal:
>
> Thank you for the support and thorough review! We are glad that you find the anti-causal setting important and the formalization based on counterfactual invariance convincing.
>
> Real-world applicability
> "There is a significant gap between performances on manually crafted tasks (synthetic setting, color MNIST) which match the underlying assumptions of the framework and on the real-world datasets (Camelyon17, PACs, VLCS). I would like to encourage the authors to address this issue and it's implications for applications on real-world datasets in more detail in the main part (especially including the important findings on real-world datasets in the main part of the paper)."
>
> In our real-world experiments, we see that the ACTIR representation enables fast domain adaptation relative to baselines. The sole exception is when we hold out the Sketch dataset in PACS, where we see ACTIR performs worse than ERM. We speculate in the paper that this is because Sketch doesn’t contain color information, but all of the training domains do. The change in invariant predictive performance is relatively minor—we think this is simply because real-world datasets don’t have shifts as dramatic as the ones used in synthetic examples.
>
> It seems that for ACTIR to yield good (adaptive) performance we need a sufficient diversity of training domains—the test domain can’t change according to a degree of freedom that’s totally distinct from how the training domains vary. For invariant prediction, it’s unclear how good an invariant predictor can be, and unclear when naive ERM will essentially achieve this performance. This is not an ACTIR-specific problem, but just a quirk of invariant learning. The fast adaptation performance on the real-world data suggests that ACTIR is indeed learning a representation that disentangles invariant and variant factors.
>
> We agree that this is an important discussion. We will use the extra page of the camera ready to bring the real-world experiments into the main paper and include the above discussion.
>
> “What conditions need the training environments to fullfill and what dataset characteristics can break the algorithm? From my point of view, it looks like diversity of the training environment is a key property?”
>
> Intuitively, it does seem that the diversity of the training datasets is key. Indeed, if the test domain varies on some factor of variation that was fixed in the training domains (e.g., color -> black in white in the sketch data) then it is hopeless to learn this factor of variation. A general characterization of the required diversity is beyond the scope of this paper but would be a fascinating direction for future work. We hope that the present paper gives a foundation for such questions in the anti-causal setting.
>
> “Would it be possible to show a similar analysis as figure 2 for one of the real-world datasets?”
>
> It’s not simple to do this because the experiment requires us to manipulate the factor Z, which is only possible when we both know what Z is and can manipulate it (neither of which holds in real-world data).
>
> "The work assumes that Xz⊥does not have a causal effect on Xz What are the implications on potential application(especially on structured datasets)?"
>
> The meaning of this assumption is that it’s possible to change the part of X not affected by Z in a manner that does not correspond to a change in the part of X affected by Z. For instance, we can change the foreground of an image without changing the background. In practice, this assumption likely doesn’t hold exactly (there are often minor interaction effects), but small violations do not present serious issues.
>
>  “During fine-tuning, are both the invariant features and the non-stable features adapted or only the non-stable features? Did the authors also consider to finetune with different learning rates, in order to allow an even more rapid adaptation?”
>
> Both invariant features and non-stable features are combined into a single representation Phi(X). During fine-tuning, we just fine-tune the last linear layer without explicitly separating the invariant and non-stable features. We did try other learning rates; this does not change the qualitative conclusion.
>
> “Is any regularization technique employed in ERM?”
>
> No regularization is employed in ERM. We don’t employ any additional regularizations for all methods to ensure a fair comparison.

---

> > ### Comment · Reviewer_UXLv · 2022-08-05
> > **Thanks for the rebuttal!**
> >
> > I would like to thank the authors for their thorough rebuttal.
> >
> > **Relation of diversity of training datasets to performance on real-world datasets**
> > I agree with the authors that "the general characterization of the required diversity is beyond the scope of this paper" and would be definitely interesting to explore in future work. However, I would like to encourage the authors to perform an ablation study on the synthetic datasets where the diversity of the training datasets can be controlled to highlight potential implications for the application on real-world datasets.
> >
> > Suggestion for future work on the role of diversity: It might be interesting to have a look at recent findings about the role of diversity in meta-learning.

---

> > > ### Author Response · Authors · 2022-08-08
> > > **Thanks for the reply!**
> > >
> > > Thanks for the reply! We'll add ablations testing the role of dataset diversity to the appendix of the camera ready. We'll also include more discussion of this in the final version.

---

### Official Review · Reviewer_ABF4 · 2022-07-12

**Rating:** 6
**Confidence:** 3
**Soundness:** 3 good
**Presentation:** 3 good
**Contribution:** 2 fair

**Summary:**

This paper proposes an invariant representation learning method across multi-domain from the perspective of the anti-causal learning process. They learn invariant and variant representation simultaneously. It is an interesting idea that considers the anti-causal direction in invariant representation learning. Experimental results show the effectiveness of the proposed method on both synthetic data and real-world datasets.

**Questions:**

1. Consider the case (Example 1) in Invariant Risk Minimization, the difficulty of anti-causal learning is that one can not have a stable prediction function if they use the effect variable to predict the cause variable (use X2 to predict Y in IRM is regarded as anti-causal learning). Therefore if the authors claim that they can achieve invariance across multi-domain, they should better give a simple example to support their motivation.
2. This algorithm can also be considered a causal learning method. if we change the direction of the edges between (y,z) and (x), the method will also hold. What makes the proposed method better for anti-causal learning should have more discussion. Causal learning and anti-causal learning should be considered separately since the independent causal mechanism(ICM) will not hold in the anti-causal direction. Note that ICM is not conditional independent, that is 'if X is cause, X is independent of the causal mechanism p(Y|X)'. If this issue can not be solved, why this method is better than IRM or other invariant representation learning where variant and invariant representation are learned simultaneously.
3. Authors should add some ablation study about the performance of the proposed method in both causal and anti-causal datasets.
4. Authors should better provide some theoretical analysis about whether they can actually learn the invariant representation, which can be regarded as the identifiability property.
5. The linear setting is limited, can the method be extended to non-linear settings?
6. Should add more state-of-art causal invariant learning methods as baselines.




**Ethics Review Area:**

["I don’t know"]

**Limitations:**

Some parts of the method should have more discussion. 1. What makes this method hold in anti-causal learning 2. Lack of theoretical analysis of identifiability 3. The linear setting is simple. More details are shown in the question part.

**Strengths And Weaknesses:**

This paper seeks an invariant learning method considering the anti-causal prediction task. They formed a causal graph to describe anti-causal prediction and learn adaption and invariant representation together. Experimental results show the effectiveness of proposed method. However, some parts of the method should have more discussion. 1. What makes this method hold in anti-causal learning 2. Lack of theoretical analysis of identifiability 3. The linear setting is simple.

---

> ### Author Response · Authors · 2022-08-01
> **Reply to reviewer**
>
> Thank you for your questions. We're glad you found the ideas interesting and the anti-causal setting significant.
>
> “Consider the case (Example 1) in Invariant Risk Minimization, the difficulty of anti-causal learning is that one can not have a stable prediction function if they use the effect variable to predict the cause variable (use X2 to predict Y in IRM is regarded as anti-causal learning). Therefore if the authors claim that they can achieve invariance across multi-domain, they should better give a simple example to support their motivation.”
>
> IRM considers a set of domain shifts defined by allowing arbitrary causal interventions on all nodes other than Y. In this case, it is not possible to make sense of invariant prediction in the anti-causal setting. However, because anti-causal problems are common in practice, in this paper, we look for an alternative formalization that enables non-trivial results. By restricting shifts so that they can only affect the latent confounder we are able to give a concrete and useful notion of invariance in the anti-causal setting. We discuss this in the first section of the literature review.
>
> “This algorithm can also be considered a causal learning method. if we change the direction of the edges between (y,z) and (x), the method will also hold. What makes the proposed method better for anti-causal learning should have more discussion. Causal learning and anti-causal learning should be considered separately since the independent causal mechanism(ICM) will not hold in the anti-causal direction. Note that ICM is not conditional independent, that is 'if X is cause, X is independent of the causal mechanism p(Y|X)'. If this issue can not be solved, why this method is better than IRM or other invariant representation learning where variant and invariant representation are learned simultaneously. “
>
> What makes this algorithm special for the anti-causal problem is the conditional independence shown in Theorem 2. If the edges are reversed, the conditional independence structure is different—the method does not hold!
>
> “Authors should add some ablation study about the performance of the proposed method in both causal and anti-causal datasets.”
>
> We have added an example to the appendix where the causal structure does not match the assumption of the method but does match the assumption of IRM, and indeed ACTIR fails and IRM succeeds. In the main paper, we show that when the assumptions hold ACTIR improves on both ERM and IRM—that is, we compare results varying only the regularization corresponding to the causal structure.
>
>  “Authors should better provide some theoretical analysis about whether they can actually learn the invariant representation, which can be regarded as the identifiability property.”
>
>  As stated in the paper, the independence criteria itself does not fully capture the counterfactual desiderata (because of the gap in the causal hierarchy). Although it would be fascinating to get an exact characterization of the gap, and an identification argument, this is beyond the scope of the paper.
>
> “The linear setting is limited, can the method be extended to non-linear settings?”
>
> We consider models that learn a non-linear representation followed by a linear map. All functions can be expressed in this form. In this sense, the method is not restricted to the linear setting. Indeed, none of our experiments are restricted to linear models. The use of linear maps on top of the learned representations is mainly for convenience—in particular, it gives a concrete and convenient formalization of fast adaptation.
>
>  “Should add more state-of-art causal invariant learning methods as baselines.”
>
> This paper serves as a proof of concept on how to learn invariant and transportable representations in anti-causal domains. The baselines we use are the methods most closely related to the ACTIR method and give a fair view of the role of anti-causal structure. In particular, ERM is the standard learning algorithm, IRM is used for invariant learning and MAML is used for adaptive learning. The ACTIR method is not aimed at improving on optimization-like aspects of these, it is instead modifying the procedure to handle anti-causal structure and learn invariant and transportable representations simultaneously.

---

> > ### Comment · Reviewer_ABF4 · 2022-08-09
> > **Thanks for authors reply!**
> >
> > I would like to thank the authors for the detailed response and revising their manuscript. Authors solve most of my concerns.
> >
> > I agree with author's response for identifiability is beyond the scope of this paper. But still encourage authors to discuss it in future version, since identifiability is a property to measure the successfulness to extract the invariant representation from observation data, which is widely discussed in previous works [1][2][3][4][5].
> >
> > [1]Ghosh S, Gresele L, von Kügelgen J, et al. On Pitfalls of Identifiability in Unsupervised Learning. A Note on:" Desiderata for Representation Learning: A Causal Perspective"[J]. arXiv preprint arXiv:2202.06844, 2022.
> > [2]Locatello F, Bauer S, Lucic M, et al. Challenging common assumptions in the unsupervised learning of disentangled representations[C]//international conference on machine learning. PMLR, 2019: 4114-4124.
> > [3]Pearl J. Causality[M]. Cambridge university press, 2009.
> > [4]Lu C, Wu Y, Hernández-Lobato J M, et al. Invariant Causal Representation Learning for Out-of-Distribution Generalization[C]//International Conference on Learning Representations. 2021.
> > [5]Wang Y, Jordan M I. Desiderata for representation learning: A causal perspective[J]. arXiv preprint arXiv:2109.03795, 2021.

---

### Meta-Review · Area_Chair_u95C · 2022-08-26

**Recommendation:** Accept
**Confidence:** Certain

**Metareview:**

In this paper, the authors propose an invariant representation learning method across multi-domain from the perspective of the anti-causal learning process. All the reviewers consider this paper is clearly written and novel.

**Award:**

Yes

---

### Decision · Program_Chairs · 2022-09-14

Accept